# Potential Biochemical Pesticide—Synthesis of Neofuranocoumarin and Inhibition the Proliferation of *Spodoptera frugiperda* Cells through Activating the Mitochondrial Pathway

**DOI:** 10.3390/toxins14100677

**Published:** 2022-09-29

**Authors:** Xuehua Shao, Zhuhong Zhang, Xuhong Qian, Lanying Wang, Yunfei Zhang, Yanping Luo

**Affiliations:** 1School of Plant Protection, Hainan University, Haikou 570228, China; 2Institute of Fruit Tree Research, Guangdong Academy of Agricultural Sciences, Key Laboratory of South Subtropical Fruit Biology and Genetic Resource Utilization, Ministry of Agriculture and Rural Affairs, Guangzhou 510640, China; 3Shanghai Key Laboratory of Chemical Biology, School of Pharmacy, East China University of Science and Technology, Shanghai 200237, China

**Keywords:** apoptosis, 2-arylfuranocoumarin, mitochondrial apoptotic pathway, photoactivated activity, Sf9 cells

## Abstract

Furanocoumarins, the secondary metabolites of plants, are considered to be natural insecticides and fungicides because they prevent the invasion of plant pathogenic microorganisms and the predation of herbivorous insects. In this study, novel 2-arylfuranocoumarin derivatives were designed to synthesize by condensation, esterification, bromination, and Wittig reaction. The results showed an excellent photosensitive activity of 2-thiophenylfuranocoumarin (I34). Cell Counting Kit-8 detected that I34 could inhibit the proliferation of *Spodoptera frugiperda* (Sf9) cells in a time- and concentration-dependent manner under ultraviolet A (UV-A) light for 3 min. The inverted microscope revealed that cells treated with I34 swelled, the membrane was ruptured, and apoptotic bodies appeared. The flow cytometry detected that I34 could induce apoptosis of Sf9 cells, increase the level of intracellular reactive oxygen species (ROS), decrease the mitochondrial membrane potential, and block cell cycle arrest in the G2/M phase. Transmission electron microscopy detected cell mitochondrial cristae damage, matrix degradation, and mitochondrial vacuolation. Further enzyme activity detection revealed that the enzyme activities of apoptosis-related proteins caspase-3 and caspase-9 increased significantly (*p* < 0.05). Finally, Western blotting analysis detected that the phosphorylation level of Akt and Bad and the expression of the apoptosis inhibitor protein Bcl-XL were inhibited, cleaved-PARP and P53 were increased, and cytochrome C was released from the mitochondria into the cytoplasm. Moreover, under UV-A irradiation, I34 promoted the increase in ROS in Sf9 cells, activated the mitochondrial apoptotic signal transduction pathway, and finally, inhibited cell proliferation. Thus, novel furanocoumarins exhibit a potential application prospect as a biochemical pesticide.

## 1. Introduction

Furanocoumarin compounds are characteristic secondary metabolites of Rutaceae and Umbelliferae, and many medicinal plants, such as *Angelicae*
*Dahuricae Radix*, *Angelicae*
*Pubescentis Radix*, *Peucedani Radix*, *Psoralea corylifolia* Linn, and so forth, which all contain these compounds [1]. They have various pharmacological activities, mainly in anti-tumor, anti-virus, anti-oxidation, anti-inflammatory, anti-pathogenic microorganisms, and insecticidal aspects [2,3,4]. In the presence of molecular oxygen, furanocoumarin molecules transfer the energy of the excited-state triplet to the ground-state triplet oxygen to generate singlet oxygen (^1^O_2_), which directly attacks the biofilm target or generates free radicals and free radical anions through electron transfer, and further reacts with oxygen to produce active oxygen. They cause strong oxidative damage to the organism, and ultimately lead to cell death through apoptotic and nonapoptotic pathways [5,6]. Furanocoumarin is a typical photosensitizer (PS) [7]. It directly attacks many biological components, such as lipids, proteins, DNA bases, and so forth, and simultaneously acts on multiple targets of insect tissue cells. It has a low risk of resistance to pests and no cross-resistance with traditional insecticides. It is eventually transformed into nonphototoxic and pollution-free residues in the environment [8,9,10]. It is a good biochemical pesticide and belongs to the category of biological pesticides. This light-activated insecticide is expected to become an effective alternative for pest control due to its lower toxicity to mammals, higher insecticidal efficiency, and environmental friendliness [11,12,13]. Previous studies showed that furanocoumarins activated by light could react with bases on DNA double-strands to form double-crosses and destroy DNA replication, inhibit the growth of a variety of cancer cells, and become a potential drug for treating cancer [14]. The effect of furanocoumarins on photoactivated pesticides was first shown by Berenbaum. The xanthotoxin inhibited the growth and development of *Spodoptera eridania* larvae [15,16]. The photoactivation of furanocoumarin pesticides was not ideal for the photosensitivity of pests due to the complex nerve and endocrine system in insects and the variability, resulting in a lack of research on its toxicological mechanism.

Furanocoumarins have a wide variety of structures. Their relative molecular mass is small, the synthesis is relatively simple, the bioavailability is high, and they have excellent development value. However, natural furanocoumarin compounds have poor biological activity against pests, therefore, based on the structure of furanocoumarin, the authors designed and synthesized a series of 2-substituted furanocoumarin derivatives to increase the π-conjugated system in the furanocoumarin structure. In the early stage of our laboratory work, it was also confirmed that 2-thiophenefurocoumarin had excellent photosensitivity to the fourth instar larvae of *Aedes aegypti* and induced midgut tissue apoptosis [17]. In this study, 39 compounds were obtained (Appendix A). Moreover, using α-terthiophene (α-T) as a positive control, the screening revealed that compound 2-thiophenylfuranocoumarin (I34) had the best biological activity against the fourth instar larvae of *A*. *aegypti*. Afterward, taking Sf9 cells as the research object, its toxicological mechanism was studied using Cell Counting Kit-8, transmission electron microscopy (TEM), flow cytometry, and Western blot analysis.

## 2. Results

### 2.1. Chemistry

In the presence of concentrated sulfuric acid, 2-methylbenzene-1,4-diol reacted with 1,3-dicarbonyl compounds 1 (ethyl acetoacetate, ethyl 2-chloroacetoacetate) to produce coumarin 2, then, at anhydrous K_2_CO_3_, aryl chloride was esterified with coumarin 2 to yield 6-ester coumarin 3, and then, compounds 3 was brominated by NBS to obtain compounds 4, which reacted with PPh3 to yield Wittig reagent 5. In the presence of K_2_CO_3_, the compounds 5 underwent Wittig reaction intramolecular at anhydrous K_2_CO_3_ to yield the title compounds I (R^1^ = H, Cl; Ar = Aryl, naphthalene, heterocycle). The route of synthesis is shown in Figure 1, and the structural formula of compound I34 is shown in Figure 1. The synthesis of target compounds requires bromination, preparation, and reaction with Wittig reagents, and these substances are sensitive to humidity. This study combined the three steps without purification and proceeded directly to the next step to obtain the target compound I so as to reduce the influence of environmental humidity and other factors.


**8-methyl-2-(thiophen-2-yl)-6H-furo [2,3-g]chromen-6-one (I34)**


White crystal, Yield: 76%, m.p.140–141 °C. 1H NMR (400 MHz, CDCl3) δ 7.68 (s, 1H), 7.56 (d, J = 3.7 Hz, 1H), 7.50–7.38 (m, 2H), 7.20–7.09 (m, 1H), 6.92 (s, 1H), 6.29 (s, 1H), 2.49 (s, 3H). 13C NMR (100 MHz, CDCl3) δ 161.15, 155.06, 152.37, 151.15, 150.26, 132.71, 132.33, 128.21, 127.34, 126.03, 117.12, 114.01, 107.60, 105.67, 100.94, 19.01. MS (EI): m/z (%) 284 ([M]+, 19), 282 (100), 254 (65), 252 (12), 226 (11), 197 (6), 127 (13). Anal. Calcd. for C16H10O3S: C, 68.07; H, 3.57; S, 11.36. Found: C, 68.05; H, 3.60; S, 11.34.

Photosensitive insecticides can be quickly activated by light, produce singlet oxygen, and induce an increase in the level of ROS in the organism, thereby acting on multiple targets of pests and making it difficult for them to become resistant [18]. These compounds are eventually transformed into nonphototoxic and nonpolluting residues in the environment, therefore, they have received extensive attention from researchers. Studies have reported that although many furanocoumarin compounds have good photosensitivity [19,20], their biological activity is poor. Based on this and the structure of furanocoumarin, this study designed and synthesized I derivatives through chemical structural modification to increase the π-conjugated system in its structure. NMR showed that the chemical shift of the 3-H atom of the title compounds I had a single peak near 6.30 ppm. The 4-methyl was also a single peak near 2.51 ppm, and a single peak near 7.09 ppm for the hydrogen of the furan ring. If the 3-H atom was replaced by chlorine, the peak near 6.30 ppm disappeared, and the 4-methyl was affected by the 3-position chlorine, and the chemical shift moved to the high field near 2.65 ppm. Mass spectrometry analysis showed that the abundance of molecular ion peaks of the title compounds were large, and some of the compounds were base peaks, which indicated that the molecular ion peaks of these compounds were stable.

### 2.2. I34 Suppressed the Proliferation of Sf9 Cells

With *Culex pipienspallens* as the target, the photosensitizing activity of the target compound was tested by ultraviolet light (365 nm, 40 W, and light distance 17 cm) at a concentration of 100 mg/L. The results showed that the target compound I had good photoactivated insecticidal activity against *C. pipienspallens* (Appendix A). In particular, the photochemical activity of compound I34 against *C. pipienspallens* exceeded 90% and no significant difference was found in the activity of the standard control α-T. Therefore, the photosensitizing compound I34 was selected as the follow-up study on the Sf9 cells.

The compounds obtained were divided into two groups, an illuminated group and a nonilluminated group, and α-T was used as a positive control. At a concentration of 100 μg/mL, the fourth instar larva of *C**. pipienspallens* was used for the activity screening. The preliminary screening results showed (Appendix A) that under light conditions, the corrected mortality of compound I34 against *C. pipienspallens* was 98.33%, which was equivalent to the photoactivity of α-T; therefore, this compound was selected for the subsequent Sf9 cells mechanism research.

The Sf9 cells treated with I34 after light activation showed significant inhibitory activity in a concentration-dependent manner compared with the dark group (*p <* 0.05). Figure 2A shows that at the treatment of Sf9 cells with I34 above 3.0 μg/mL, the cell inhibition rate of the light-activated group was more than 60%, while that of the dark group was less than 10%. When the treatment concentration of I34 was 10.0 μg/mL, the cell inhibition rate of the dark group was 10.45% and that of the light-activated group reached 77.46%. Under light activation conditions, the inhibitory intermediate concentration (half-maximal inhibitory concentration) of the I34 compound for 24, 48, and 72 h was 6.333, 2.424, and 1.620 μg/mL, respectively. The photoactivation toxicity of compound I34 to Sf9 cells was both time and concentration dependent (Figure 2B).

In this study, an inverted microscope was used to observe the effect of light-activated I34 on the morphology of Sf9 cells. Sf9 cells treated with I34 (2.5 μg/mL) for 0 h had a full external morphology, irregular round shape, and a better adherent growth state. When the cells were treated for 3 and 6 h, most of the cells became rounded and swelled. However, after 12 h, the number of swelled cells increased significantly, and cell proliferation was inhibited. When the cells were treated for 24 h, the cells could not grow adherently, the cell membrane was ruptured, and apoptotic bodies appeared. After 48 h, the cells swelled and ruptured, the contents overflowed, apoptotic bodies increased, and a large number of cells died (Figure 2D).

### 2.3. I34 Arrested Cell Cycle

Flow cytometry was performed to observe the distribution of the cell cycle to determine the role of I34 in the cell cycle of Sf9 cells. The Sf9 cells were treated with light-activated I34 at a concentration of 2.5 μg/mL for 3, 6, 12, 24, and 48 h. As shown in Figure 2C,E, the treated cells significantly decreased the percentage of cells in the G1/G0 phase compared with 0 h, while the number of cells in the G2/M phase increased significantly. Among them, the number of cells in the G0/G1 phase proportion decreased from 28.49% to 12.27% after 48 h, while the number of cells in the G2/M phase proportion increased from 29.87% to 53.53%. These data indicated that photoactivated I34 regulated Sf9 cell proliferation by inducing cell cycle arrest in the G2/M phase. The cell cycle was the main regulatory mechanism that controlled cell growth. The cells were always in a continuous cell cycle from G1, S, G2 to M phase due to the continuous proliferation of normal cells [21]. At each stage of the cell cycle, the cells performed important physiological activities, such as DNA replication, protein synthesis, and cell division [22]. Among them, the G2/M phase underwent cell division after DNA replication was completed and initiated the subsequent cell cycle process. However, most of the sf9 cells treated with I34 stayed in the G2/M phase, and the cells were unable to undergo normal mitosis, thereby inhibiting cell proliferation.

### 2.4. Effects of I34 on the Intracellular ROS Level in Sf9 Cells

In this study, the cells were stained with 2′-7′dichlorofluorescin diacetate (DCFH-DA) and 2′-7′dichlorofluorescein (DCF) fluorescence and measured using a flow cytometer to examine the production of ROS in Sf9 cells. Figure 3A shows that the DCF fluorescence peaks all moved to the right along the abscissa when the Sf9 cells were exposed to 2.5 μg/mL of I34 for 3, 6, 9, and 12 h, leading to a remarkable increase in the intracellular ROS compared with 0 h, respectively, and with an increased ROS level represented by the DCF intensity. Through statistical analysis, the intracellular ROS level of Sf9 cells treated with I34 for 3, 6, 9, and 12 h was determined to be 1.67, 1.43, 1.21, and 1.11 times of that at 0 h, respectively. The ROS were found to peak within 3 h in the Sf9 cells after the I34 treatment. Then, at 6 h, when the DNA damage was intense, the ROS level decreased (Figure 3B). Furthermore, the changes in the fluorescence intensity of DCF were observed using a confocal microscope (Figure 3C), and the ROS level was found to be the highest within 3 h, which was consistent with the results using flow cytometry. ROS are chemically active molecules produced by a reduction in molecular oxygen, including some oxygen-containing free radicals, peroxides, singlet oxygen molecules, and so on [23,24,25]. Under normal circumstances, organisms can maintain the balance of ROS metabolism in the body [26]. Photosensitivity reactions are often accompanied by the generation of free radicals and the transfer of free radical electrons, and induce the increase in intracellular ROS [27,28]. If these excessively generated ROS cannot be eliminated, they cause cell oxidative stress damage. This study found that I34 photoactivation promoted a significant increase in the level of ROS in Sf9 cells, and the cumulative amount of ROS in the cells was the highest after treatment for 3 h.

### 2.5. Effect of I34 on the Ultrastructure in Sf9 Cells

The Sf9 cells were treated with 2.5 μg/mL of I34 for 0, 12, 24, and 36 h, and then the ultrastructural changes of the cells were observed using a TEM. As shown in Figure 3D, when the Sf9 cells were treated with I34 for 0 h, the cell morphology was complete, the nucleus was larger, the nuclear membrane was clear, the chromatin was gathered in the middle of the nucleus, and the mitochondria had clear outlines and complete cristae. After 12 h of treatment, the nucleus volume became smaller, the chromatin was scattered, some mitochondria were damaged and dissolved, and the mitochondrial cristae structure was blurred. After 24 h of treatment, the mitochondrial cristae were severely damaged, and the cell lysosomes were increased. After 36 h of treatment, the chromatin in the nucleus condensed into clumps and distributed around the nuclear membrane, the nuclear membrane swelled and ruptured, the mitochondrial cristae disappeared, and the mitochondria appeared vacuolized. Therefore, I34 could severely damage the mitochondrial structure of Sf9 cells. The nuclear membrane of the Sf9 cells swelled and ruptured, and the mitochondrial cristae disappeared and showed vacuolation, which was detected by TEM. I34 caused the mitochondrial structure damage and dysfunction of Sf9 cells, which corresponded to the increased levels of ROS detected. Excessive accumulation of ROS could not only lead to oxidative stress but also induce cell function loss, cell cycle arrest, and even cell apoptosis. This study first found that I34 induced apoptosis of Sf9 cells, using flow cytometry to verify this conclusion, and observed that it was time-dependent. Further, using a laser confocal microscope, it was detected that the light-activated I34 could react with the bases on the DNA double-strand to break it (Figure 4). This destroyed the ability of DNA to replicate and inhibited the proliferation of the Sf9 cells, which was consistent with the results of previous studies [29].

### 2.6. I34 Induced Sf9 Cell Apoptosis

The cells were pretreated for different times (0, 3, 6, 12, 24, and 48 h) to observe whether I34 was capable of inducing apoptosis. After the treatment, the cells were stained with PI and fluorescein isothiocyanate (FITC)-labeled Annexin V (AV-FITC) to analyze the percentage of apoptotic cells using flow cytometry. The lower left quadrant indicates normal cells, the upper left quadrant indicates necrotic cells, the lower right quadrant indicates early apoptotic cells, and the upper right quadrant indicates late apoptotic cells (Figure 4A). Figure 4C shows that the total (early + late) apoptotic cells drastically enhanced from 0 h to 3 h and 6 h, while the normal cells sharply decreased. At 48 h, the normal cells decreased from (89.42 ± 0.51) % to (12.12 ± 0.91) %, while the apoptotic cells increased from (5.96 ± 0.48) % to (58.30 ± 1.40) % (Figure 4A,C). Light-activated I34 could induce apoptosis of Sf9 cells, and the apoptotic rate positively correlated with the treatment time.

ROS could result in a free radical attack of membrane phospholipids, leading to a loss of mitochondrial membrane potential. Mitochondrial depolarization was considered an irreversible step in the apoptotic process [30,31]. Therefore, the ability of Sf9 cell mitochondria to maintain membrane potential in the six groups was measured using flow cytometry techniques. The I34 treatment for 3, 6, 12, 24, and 48 h could cause serious depolarization of the inner mitochondrial membrane in Sf9 cells compared with that at 0 h. Figure 4B,D shows that the ratio of red and green fluorescence in the cells treated for 3 h was significantly lower than that at 0 h, and the longer the treatment time, the more significant the decrease in membrane potential. This conclusion was consistent with the results of the TEM.

A marker of apoptosis was scored by performing a terminal deoxynucleotidyl transferase dUTP nick end labeling (TUNEL) assay that measured DNA fragmentation (compared with that at 0 h, *p* < 0.05), a characteristic feature of apoptosis. Figure 4F shows that at 0 h, no obvious green fluorescence was observed, and the cell condition was normal. At 6 h, terminal deoxynucleotide transferase added labeled nucleotides to the 3′ termini at double-stranded breaks in the fragmented DNA. The whole cell showed a lighter green fluorescence, and cell apoptosis occurred. At 12 h, more green fluorescent spots appeared in the cell nucleus, DNA break sites increased, I34 caused a significant increase in the number of TUNEL-positive cells (*p* < 0.05 compared with that at 0 h), the number of apoptotic cells increased, and the number of normal cells decreased. At 24 h, green fluorescent spots still occurred at the DNA breaks in the cell nucleus, and apoptosis still existed. However, the green fluorescence intensity of the whole cell decreased due to severe damage to the cell membrane. Apoptosis is mainly achieved through two signaling pathways, including the intracellular (mitochondrial-mediated) pathway and the extracellular (death receptor-mediated) pathway [32]. The two main features of the intracellular apoptotic pathway are the loss of mitochondrial membrane potential, and the release of cytochrome C. Cytochrome C is positively charged on the surface, and is usually loosely bound to cardiolipin, rich in unsaturated fatty acids, to fix the outer surface of the inner mitochondrial membrane [33]. Excessive ROS can cause oxidative damage to cardiolipin, causing cytochrome C to be dissociated and released into the cytoplasm, causing mitochondrial depolarization [34,35]. The results of this study found that the mitochondrial membrane potential of Sf9 cells treated with I34 for 3 h decreased in a time-dependent manner (Figure 4).

### 2.7. I34 Induced Sf9 Cell Apoptosis through the Mitochondrial Signaling Pathway

The Sf9 cells were treated with 2.5 μg/mL of I34 for 0, 3, 6, 12, 24, and 48 h, and the enzyme activities of caspase-3 and caspase-9 were measured. The results (Figure 5A) showed that the enzyme activities of caspase-3 and caspase-9 in the cells treated for 3, 6, 12, and 24 h were significantly higher than those at 0 h. Among them, the enzyme activity was highest at 12 h, and the enzyme activity of caspase-3 and caspase-9 were 2.22 and 3.05 times of those at 0 h, respectively. In addition, the enzyme activities of caspase-3 and caspase-9 decreased significantly when the Sf9 cells were treated with I34 for 48 h, which might be due to the large number of cell deaths.

The mRNA expression of apoptosis-related genes was detected by quantitative reverse transcriptase–polymerase chain reaction (qRT-PCR). Figure 5B shows that the expression level of the *Sf-Caspase-1* gene was significantly upregulated, of which the expression level of the gene at 12 h was upregulated by 7.06 times compared with that at 0 h. The expression level of the *Sf-Caspase-5* gene was upregulated by 3.05 times at 24 h compared with that at 0 h. After the I34 treatment of the Sf9 cells for 6, 12, and 24 h, the expression level of the pro-apoptotic factor *Sf-Apaf1* was upregulated by 3.08, 3.77, and 3.31 times compared with that at 0 h. The expression levels of *Sf-Cytochrome C* and *Sf-p53* genes were also significantly upregulated after treatment compared with those at 0 h; they were upregulated 3.41 times and 3.58 times, respectively. The expression level of the *Sf-Cathepsin L* gene was significantly upregulated (*p* < 0.05), among which the expression level of the gene at 12 h was upregulated by 2.99 times compared with that at 0 h. The expression levels of the key genes *Sf-Caspase-1*, *Sf-Caspase-5*, *Sf-Apaf1*, *Sf-Cytochrome C*, *Sf-p53*, and *Sf-Cathepsin L* in the apoptotic pathway were all significantly upregulated (*p* < 0.05) in Sf9 cells treated with I34 for 6, 12, and 24 h; however, the six genes showed a downward trend at 48 h.

The Western blot analysis was used to verify the expression of apoptosis-related proteins of I34 treatment for different times. Figure 5C,D show that the expression level of phospho-protein kinase A (pAkt) protein was significantly downregulated at 6 h and that of the pro-apoptotic protein Bcl-XL was significantly upregulated at 3 h, while the expression level of the anti-apoptotic protein phospho-Bad (Ser112) was significantly downregulated. After 12, 24, and 48 h, the expression level of apoptosis-indicating protein cleaved-poly (ADP-ribose) polymerase (PARP) was significantly increased, cytochrome C was released into the cytoplasm, and the P53 protein was also increased significantly with the prolongation of the treatment time (*p* < 0.05). The TEM also showed that the mitochondrial structure was seriously damaged (Figure 3). In the mitochondrial-mediated apoptosis pathway, caspase-9 was the apoptosis-initiating factor in the cascade reaction. It formed a complex with the accessory protein cytochrome C, Apaf-1 was cleaved and activated, and this further activated the apoptosis executive factor of caspase-3 downstream of the cascade. The results indicated that caspase-9 and caspase-3 enzyme activities of Sf9 cells after I34 treatment first increased and then decreased. The caspase-9 and caspase-3 enzyme activities were highest in the cells treated for 12 h. qRT-PCR detected that the expression levels of *Sf-Caspase-1*, *Sf-Caspase-5*, *Sf-Apaf1*, *Sf-Cytochrome C*, *Sf-p53*, and *Sf-Cathepsin L* genes were upregulated. Further, I34 induced Sf9 cell apoptosis through the mitochondrial apoptosis signaling pathway.

In cell apoptosis, the activation of Akt directly inhibits apoptosis by phosphorylation and deactivation of pro-apoptotic factors such as *Bad*, *caspase-9*, and transcription factor *FOXO3a*, or by inhibiting the conformational changes of the pro-apoptotic protein Bax [36,37]. PARP participates in cell apoptosis through the cleavage of caspase-3, and the cleavage of PARP is considered to be one of the indicators of caspase-3 activation. Members of the Bcl-2 family play a regulatory role in mitochondrial dysfunction. Bcl-XL can compete with cytochrome C for the binding site of Apaf-1, so that Apaf-1 does not participate in the activation of pro-caspase-9, thereby inhibiting cell apoptosis. Nonphosphorylated Bad combines with the anti-apoptotic proteins Bcl-2 and Bcl-XL to form a complex, and this transfers the pro-apoptotic factor Bax [38]. In this study, Western blot analysis detected that the expression levels of pAkt (Ser473) and the anti-apoptotic protein phospho-Bad (Ser112) of Sf9 cells treated with I34 were significantly downregulated, and those of cleaved-PARP and P53 were upregulated. In summary, I34 induced Sf9 cell apoptosis through the caspase-dependent mitochondrial pathway, leading to inhibition of its proliferation.

## 3. Conclusions

In this study, a novel 2-arylfuranocoumarin (I34) derivative was obtained through condensation, esterification, bromination, and Wittig reaction. According to the biological evaluation, I34 showed excellent photosensitive activity. Cell Counting Kit-8 detected that I34 could inhibit the proliferation of *S**. frugiperda* (Sf9) cells in a time- and concentration-dependent manner under ultraviolet A (UV-A) light for 3 h. The inverted microscope revealed that cells treated with I34 swelled, the membrane was ruptured, and apoptotic bodies appeared. The flow cytometry detected that I34 could induce apoptosis of Sf9 cells, increase the level of intracellular reactive oxygen species (ROS), decrease the mitochondrial membrane potential, and block cell cycle arrest in the G2/M phase. Transmission electron microscopy (TEM) detected cell mitochondrial cristae damage, matrix degradation, and mitochondrial vacuolation. Further enzyme activity detection revealed that the enzyme activities of the apoptosis-related proteins caspase-3 and caspase-9 increased significantly (*p* < 0.05). Finally, Western blot analysis detected that the phosphorylation level of Akt and Bad and the expression of apoptosis inhibitor protein Bcl-XL were inhibited, cleaved-PARP and P53 were increased, and cytochrome C was released from the mitochondria into the cytoplasm. Moreover, under UV-A irradiation, I34 promoted an increase in ROS in Sf9 cells, activated the mitochondrial apoptotic signal transduction pathway, and finally, inhibited cell proliferation. Thus, novel furanocoumarins have a great potential to be exploited in the development of new biochemical pesticides.

## 4. Materials and Methods

### 4.1. Experimental Materials and Equipment

Sf9 cells (*Spodoptera frug**iperda* ovarian cell line) were cultured by the Laboratory of Resources and Environment, Fruit Research Institute, Guangdong Academy of Agricultural Sciences. The cells were maintained in SIM SF Expression Medium (MSF1) supplemented with 10% fetal bovine serum at 27.0 °C.

The SIM SF Expression Medium (MSF1) was purchased from Sino Biotechnology Company (Sino Biological, Beijing, China). The Fetal Bovine Serum (40130ES76) was purchased from Yeasen Biotechnology Company (Yeasen, Xi’an, China). The Cell Counting Kit-8 (CCK-8) (C0038), DNA Content Quantitation Assay (Cell Cycle) (CA1510), Annexin V-FITC Apoptosis Detection Kit (CA1020), Mitochondrial Membrane Potential Assay Kit with JC-1(M8650), TUNEL Apoptosis Assay Kit (T2190) and Mounting Medium, antifading (with DAPI) (S2110) were purchased from Solarbio Technology Company (Solarbio, Beijing, China). The Caspase 3 Activity Assay Kit (C1116), Caspase 9 Activity Assay Kit (C1158), α-Tubulin antibody (AT819), HRP-labeled Goat Anti-Rabbit IgG(H+L) (A0208) and HRP-labeled Goat Anti-Mouse IgG(H+L) (A0216) were purchased from Beyotime Biotechnology Company (Beyotime, Beijing, China). Phosphor-Akt (Ser473) rabbit mAb (#4060), Phosphor-Bad (Ser112) rabbit mAb (#5284), Cleaved PARP (Asp214) rabbitAb (#5625), Akt antibody (#9272), p53 rabbit mAb (#2527), Cytochrome C antibody (#4272) and Bcl-XL antibody (#2764) were purchased from Cell Signaling Technology company in the USA. Others came from commercially available analytical or chemical reagents and medicines.

Melting points (m. p.) were measured on the X6 micromelting-point apparatus (China) and are uncorrected. The equipment included an AV 400 NMR spectrometer (Bruker, Swiss; CDCl3 was used as a solvent), a Trace MS 2000 GS-MS (Thermo Finnigan, San Jose, CA, USA), a Vario EL III element analyzer (Elementar, Langenselbold, Germany), and a mixer mill MM400 frozen (Retsch, Haan, Germany).

### 4.2. Synthesis of Furan [2,3-g] Coumarin I

The synthesis of the compounds in the manuscript refers to the previous research of the College of Plant Protection, Hainan University [17,39].

### 4.3. Cytotoxicity Assays

Inhibition of Sf9 cell proliferation by I34 was measured by the Cell Counting Kit-8 (CCK-8) assays. Briefly, 1 × 10^5^ cells were seeded in 96-well plate for each well. There were six repeats for each system. After 24 h incubation, the cells were treated with I34 (0.5, 1.0, 2.0, 3.0, 5.0, 7.0 and 10.0 μg/mL) for 24 h, 48 h and 72 h. Then, 10 μL CCK-8 was added to the 96-well plate for each well and incubated for 3 h. The absorbance at 450 nm was measured by a Microplate spectrophotometer (ELx-800, Biotec Instruments, Winooski, VT, USA). With reference to Huang’s lighting method, the experiment was divided into a dark group and a light group. The light group was cultured for 3 h, and two UV lamps with a spectral peak of 365 nm and 40 W were used as the light source. The ultraviolet intensity was 25 mW/cm^2^, the light distance was 17 cm, the light time was 3 min, and the dark group was not subjected to light treatment.

### 4.4. Cell Morphology Detection

The Sf9 cells in the logarithmic growth phase were discarded from the cell culture medium and replaced with fresh cell culture medium containing 2.5 μg/mL of I34. The light treatment of the cells was the same as Section 2.3. After the cells were treated with I34 for 0 h, 3 h, 6 h, 12 h, 24 h and 48 h, they were observed under an inverted microscope and photographed to record the changes in cell morphology.

### 4.5. Cell Cycle Arrest and Apoptosis Assay

Sf9 cells were seeded in 6-well plates at a density of 4 × 10^5^ cells/well. After the treatment, the wells were washed three times with phosphate buffer saline (PBS), 70% cold ethanol was added and kept at 4 °C for 2 h. PBS was used to wash the cells again and the cells were resuspended. The cells were stained with 25 μL PI (50 μg/mL) and incubated with 10 μL RNase (50 μg/mL) at 37 °C for 30 min. Final fluorescence detection was performed using a Flow Cytometer.

An Annexin V-FITC Apoptosis Detection Kit (Beyotime) was used to detect cell apoptosis according to the manufacturer’s instructions. Sf9 cells were plated in 6-well plates with 1×10^5^ cells per well and treated with 2.5 μg/mL I34 for 0 h, 3 h, 6 h, 12 h, 24 h and 48 h at 27 °C following adherence. Floating and adherent cells were collected and resuspended in 100 µL binding buffer, which containing 5 µL FITC-conjugated annexin-V and 5 µL PI, then incubated for 15 min at room temperature. After which, 400 µL binding buffer was added to the suspension and then immediately analyzed by flow cytometry (Accuri C6, BD Biosciences, San Jose, CA, USA) and observed under a fluorescence microscope (DMRA, Leica Microsystems, Wetzlar, Germany).

### 4.6. Reactive Oxygen Species (ROS) Assay

A ROS Assay kit (Beyotime) was used to determine the level of intracellular ROS. Considering that the production of ROS must be earlier than the presence of cell apoptosis and cell cycle arrest, 0 h, 3 h, 6 h, 9 h and 12 h were selected as the treatment time in this assay. The following steps were performed according to the manufacturer’s protocol. Each well was washed three times with 1 × PBS and incubated with 10 μM dichloro-dihydro-fluorescein diacetate (DCFH-DA) at 27 °C for 30 min in the dark. After incubation, the cells were washed with PBS and observed under the fluorescence microscope (DMRA, Leica Microsystems, Wetzlar, Germany). To quantify ROS levels, the mean fluorescence intensity (MFI) was detected with flow cytometry (Accuri C6, BD Biosciences, San Jose, CA, USA) and analyzed using FlowJo V10 software.

### 4.7. Mitochondrial Membrane Potential Assay

Sf9 cells were seeded in 6-well plates at a density of 4 × 10^5^ cells/well. The cells were treated with 2.5 μg/mL I34 for 0 h, 3 h, 6 h, 12 h, 24 h and 48 h. The mitochondrial membrane potential (MMP) was measured with a mitochondrial membrane potential assay kit with JC-1 (Beyotime) according to the manufacturer’s instruction. After JC-1 staining, the cells were also resuspended with a culture medium and analyzed by flow cytometry (Accuri C6, BD Biosciences, San Jose, CA, USA).

### 4.8. Transmission Electron Microscopy (TEM) Analysis

The cells were seeded in 6-well plates at a density of 4 × 10^5^ cells/well for 48 h. After treatment with 2.5 μg/mL I34 for different times (0, 12, 24 or 36 h), the cells were fixed in 1% osmium tetraoxide for 4 h and embedded in Spur Resin according to the manufacturer’s instructions. Once fixed, the specimens were cut to 60 nm thickness and stained with 0.5% uranyl acetate and lead citrate. Finally, the cells were observed using a transmission electron microscope (Tecnai 12, FEI Co., Eindhoven, The Netherlands).

### 4.9. TUNEL Staining

Sf9 cells were plated in 6-well plates with 1 × 10^5^ cells per well and treated with 2.5 μg/mL I34 for 0 h, 6 h, 12 h and 24 h at 27 °C. After incubation, the cells were washed three times with PBS and fixed with 4% paraformaldehyde for 20 min. The fixative (paraformaldehyde) was then discarded, and the cells were washed with pre-cooled PBS three times, each time for 5 min at room temperature. After washing, the sections were incubated with the TUNEL reaction solution, containing terminal deoxynucleotidyl transferase and fluorescein-labeled dUTP, at 37 °C for 1h in the dark. Then, DAPI was added for staining. Finally, the sections were sealed with anti-fluorescence quenching sealed tablets after washing three times with phosphate-buffered saline (PBS). The number of TUNEL-positive cells was counted on 10 randomly selected ×100 fields for each section using a fluorescence microscope (DMRA, Leica Microsystems, Wetzlar, Germany). Apoptotic index (AI) = number of positive cells/total number of cells ×100%.

### 4.10. Caspase-3 and Caspase-9 Activity Assay

Caspase-3 and caspase-9 activities were analyzed using the caspase-3 and caspase-9 activity assay kits (Beyotime) according to the manufacturer’s protocol. The cells were processed in the same way as 2.7 and were lysed by lysis buffer (50 μL) for 15 min on ice. Cell lysates were centrifuged at 12,000× *g* for 10 min at 4 °C, the supernatants were collected, and protein concentration was determined by the Bradford’s method. The reaction was added to the reaction buffer containing Ac-DEVD-pNA (10 μL) and Ac-LEHD-pNA (10 μL) and incubated for 2 h at 37 °C. Then, the samples were measured using the MicroplateReader (SPECTRAMAX 190, Molecular Devices, Sunnyvale, CA, USA) with a wavelength of 405 nm. The caspase-3 and caspase-9 activities were measured as a fold of enzyme activity in comparison with the control. All the experiments were conducted in triplicate.

### 4.11. Quantitative Real-Time PCR (qRT-PCR)

Total RNA was extracted by the Total RNA extraction kit, then the RNA (1 μg) was reverse-transcribed into cDNA in a 20-μL reaction system using a High Capacity cDNA Reverse Transcription Kit (Applied Biosystems, Waltham, MA, USA). Real-time PCR was performed using a Power SYBR^®^ Green PCR Master Mix and a 7500 FAST Real-time PCR System (Applied Biosystems ABI, Carlsbad, CA, USA). *Sf-GAPDH* was used as an internal control for the measurement of mRNA levels. The primer sequences are shown in Table 1. Each treatment was repeated 3 times, and the 2^-ΔΔCt^ method was used to calculate the gene expression of Sf9 cells under different treatment times [40].

### 4.12. Western Blot

The total protein was extracted with RIPA lysis buffer, lysed on ice for 5 min. The specimens were left to rest for 30 min and centrifuged for 10 min at 4 °C at 12,000× *g*, then the supernatant was collected and stored at −20 °C. The BSA protein assay was used to quantify the protein levels. The proteins were extracted from the cells and 20 µg aliquots were subjected to SDS-PAGE and transferred onto PVDF membranes. The transfer was initially performed for 40 min and subsequently adjusted to 110 V for 90 min. The membrane was incubated with the following primary antibodies, which were all diluted to 1:500 in TBST. Incubations were maintained at 4 °C overnight before rinsing with TBST 3 times, each time for 10 min. The membrane was incubated with goat or mouse anti-rabbit anti-second antibodies at room temperature for 2 h. The protein bands were analyzed using the Image Quant LAS GEL imaging system (4000 biomolecular imager, GE Healthcare, Little Chalfont, UK). Image J software was used to analyze the band gray values and calculate the relative protein expression levels.

### 4.13. Statistical Analysis

Statistical analyses were performed using SPSS 23.0 software (SPSS, Chicago, IL, USA). The data are displayed as mean values ± SD. In vitro and in vivo experiments were assessed in at least three independent experiments. Student’s *t*-test was used to compare the differences between the two groups, and multiple comparisons of means were conducted using one-way analysis of variance, followed by Tukey’s Honestly Significant Difference for pair–wise comparisons. The significance level was at α = 0.05.

## Data Availability

The data presented in this study are available on request from the corresponding author.

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
