# Peer review of "Potential Biochemical Pesticide—Synthesis of Neofuranocoumarin and Inhibition the Proliferation of Spodoptera frugiperda Cells through Activating the Mitochondrial Pathway"

_toxins, 2022, doi:10.3390/toxins14100677_

Round 1

Reviewer 1 Report

Major concern:

Lines 249-535: change these paragraphs as an Appendix.

Minor concerns:

Line 9: to synthesize

Line 16: delete (TEM).

Line 19: western blotting

Line 21, line 808: remove “(Cyt C)”

Line 24: delete “will” in furanocoumarinswill

Line 30-69: apply square brackets for the numerals for the references through the context.

Line 72-73: provide the manufacturer name for Sf9 cells instead of the donator.

Line 88: cytochrome C

Line 99: of College

Line 101: the Cell

Line 108: for 3 h,

Line 132: Inc.).

Line 106: cite the reference 17 here.

Line 111: 2.5 μg/mLof I34.

Line 129-130, line 166: change “propidium iodide (PI)” to PI.

Line 135, line 149: (Beyotime) is enough.

Line 183: were conducted in triplicates.

Line 195: 12,000×g.

Line 212: followed by Tukey's Honestly Significant Difference for pair - wise comparisons.

Lines 212-213: The significance level was at α = 0.05.

Line 240: Iy?

Line 537: Culex pipienspallens

Line 546: C. pipienspallens

Line 551: activity in

Line 553: at the treatment of Sf9 cells with I34 above 3.0 μg/mL,

Line 559-560: was both time and concentration dependent.

Title of x-axis of Figure 2A and B: concentration of LX76?.

Line 599: in Sf9 cells

Line 608: peak within, …., at 6 h,

Line 664: Figure 4C shows

Lines 698-699: (compared with that at 0 h, P < 0.05),

References: the style of references was not in accord with Toxins.

Author Response

Re: Manuscript ID: toxins-1918906

Title: Potential Biochemical Pesticide Synthesis of Neofuranocoumarin and

Inhibition the Proliferation of Spodoptera frugiperda cells through activating

the mitochondrial pathway

Dear Reviewer:

Thank you for your careful and responsible review of our manuscripts for helping us improve our manuscript in both scientific and format aspects. We provide this letter to explain, point by point, the details of our revisions in the manuscript and our responses to the comments as follows. In order to make the changes easily viewable for you, we used revision mode in the revised manuscript. Besides, we have carefully checked through the whole manuscript and corrected some grammar mistakes. We hope the revised paper would satisfy you.

  1. Major concern: Lines 249-535: change these paragraphs as an Appendix.

Reply:We are very grateful to the reviewer for the suggestion. Because the I34 was extensively tested, we leave only the data for I34, other compounds have been placed in the supplementary material.

  1. Line 9: to synthesize

Reply:We apologize for this mistake, we have corrected it in the text, Details of the changes can be found in line 10.

  1. Line 16: delete (TEM).

Reply:Thank you for your suggestion, we have deleted (TEM) in the text, Details of the changes can be found in line 18.

  1. Line 19: western blotting

Reply:Thank you for your suggestion, we corrected this grammatical error in the draft and the changes can be found in line 21.

  1. Line 21, line 808: remove “(Cyt C)”

Reply:Thank you for your suggestion, we have removed “(Cyt C)”.

  1. Line 24: delete “will” in furanocoumarinswill

Reply: Thank you for your careful review. we have corrected it in the text, Details of the changes can be found in line 26.

  1. Line 30-69: apply square brackets for the numerals for the references through the context.

Reply: We apologize for the format problem in the original manuscript. The references in the original manuscript we uploaded are boxed with square brackets. Nevertheless, we again made changes to the references in the manuscript.

  1. Line 72-73: provide the manufacturer name for Sf9 cells instead of the donator.

Reply: Thank you for your careful review. We provide the manufacturer name for Sf9 cells in line 74.

  1. Line 88: cytochrome C

Reply: Thank you for your careful review. We have capitalized the letter C, see in line 91.

  1. Line 99: of College

Reply: We have added Spaces, and we have read through the text to correct such errors.

  1. Line 101: the Cell

Reply: Thank you for your careful review. We have added spaces and we have read through the text to correct such errors.

  1. Line 108: for 3 h,

Reply: Thank you for your careful review. We have changed “hours” to “h”.

  1. Line 132: Inc.).

Reply: Thank you for your careful review, we have changed in line 135.

  1. Line 106: cite the reference 17 here.

Reply: Thank you for your suggestion, we have revised in the text.

  1. Line 111: 2.5 μg/mL of I34.

Reply: Thank you for your careful review. We have added spaces and we have read through the text to correct such errors, see in line 116.

  1. Line 129-130, line 166: change “propidium iodide (PI)” to PI.

Reply: Thank you for your careful review. Change “propidium iodide (PI)” to PI.

  1. Line 135, line 149: (Beyotime) is enough.

Reply: Thank you. We have changed “(Beyotime Institute of Biotechnology)” to “(Beyotime)”.

  1. Line 183: were conducted in triplicates.

Reply: Thank you for your careful review. We have corrected this grammatical error.

  1. Line 195: 12,000×g.

Reply: Thank you for your careful review. This writing error has been corrected.

  1. Line 212: followed by Tukey's Honestly Significant Difference for pair - wise comparisons.

Reply: Thank you for your careful review, we have corrected this grammatical error, see in line 212.

  1. Lines 212-213: The significance level was at α = 0.05.

Reply: Thank you for your careful review. The significance level was at P≤ 0.05.  We have corrected this grammatical error; see in lines 214-215.

  1. Line 240: Iy?

Reply: Thank you for your careful review. After our inspection, we found that this was a writing error, and we deleted “y”, see in line 64.

  1. Line 537: Culex pipienspallens

Reply: Thank you for your careful review. We have Changed “C. pipienspallensas”to “Culex pipienspallens”.

  1. Line 546: C. pipienspallens

Reply: Thank you for your careful review. We have Changed “Culex pipienspallens”to “C. pipienspallensas”.

  1. Line 551: activity in

Reply: Thank you for your careful review. We have added spaces and we have read through the text to correct such errors.

  1. Line 553: at the treatment of Sf9 cells with I34 above 3.0 μg/mL

Reply: Thank you for your careful review. We have corrected this grammatical error.

  1. Line 559-560: was both time and concentration dependent.

Reply: Thank you for your careful review. We have corrected this grammatical error.

  1. Title of x-axis of Figure 2A and B: concentration of LX76?

Reply: Thank you for your careful review. We've already confirmed that the   titles of x-axis of Figure 2A and B were concentrations of LX76.

  1. Line 599: in Sf9 cells

Reply: We have added spaces and we have read through the text to correct such errors.

  1. Line 608: peak within, …., at 6 h,

Reply: Thank you for your careful review. We have added spaces and we have read through the text to correct such errors.

  1. Line 664: Figure 4C shows

Reply: Thank you for your careful review. We have added spaces and we have read through the text to correct such errors, see in line 393.

  1. Lines 698-699: (compared with that at 0 h, P < 0.05),

Reply: We apologize for the incorrect formatting, We have corrected this grammatical error, see in line 411.

  1. References: the style of references was not in accord with Toxins.

Reply: We apologize for the incorrect formatting of the references and we have revised again in the text.

We would like to thank you for allowing us to resubmit a revised copy of the manuscript and we highly appreciate your time and review. Thank you again for your contribution to our manuscript.

We are looking forward to hearing from you soon.

Kind Regards,

Yanping Luo

Reviewer 2 Report

Dear author, I have carefully reviewed the manuscript “Potential Biochemical Pesticide - Synthesis of Neofuranocoumarin and Inhibition the Proliferation of Spodoptera frugiperda cells through activating the mitochondrial pathway.” In this manuscript, the authors related the synthesis of a new furanocoumarin and evaluated its potential as a biochemical pesticide. The manuscript is well-written and a pleasure to read. The review believes that it meets the publication requirements if the author can provide more information on the following questions and revise their manuscript accordingly

Suggestions to make the text easier to read:

- Incorporate the notes in the text legend.

- Although the authors have synthesized 39 compounds, only the I34 was extensively tested. Leave only the RMN data of I34 in the manuscript; the RMN of the other compounds could be added as supplementary material with their structures.

- Make an extensive spell check, e.g. lines 24, 61, 99, 101, 132, and so on.

Minor points.

Line 72: Correct spelling frugiperda

Line 122: Provide the concentration of PI and RNAse.

Line 537: First time that appear C. pipienspallens change by Culex pipienspallens.

Line 538: In materials and methods, you wrote that the light distance is 17 cm instead 15 cm. Which is correct?

Line 592: (C) is a statistic analysis of figure 2E.

Line 645: Put the scale bar in figure 3D.

Line 682-683: Confocal microscopy in this magnification is unsuitable for showing morphological changes. I Suggest removing this phrase, as well as the figure 4E. If you wish can put the figure as supplementary material.

Line 713: Suggest including the statistic analysis of figure 4F.

Line 798: The authors wrote that they used UV light for 3 h, but in the materials and methods, the time was 3 min (line 110) which is a huge difference. Please correct.

Author Response

Re: Manuscript ID: toxins-1918906

Title: Potential Biochemical Pesticide Synthesis of Neofuranocoumarin and

Inhibition the Proliferation of Spodoptera frugiperda cells through activating

the mitochondrial pathway

We really appreciate your efforts in reviewing our manuscript during this unprecedented and challenging time. We wish good health to you, your family, and community. We gratefully thank you for you making constructive remarks and useful suggestion, which has significantly raised the quality of the manuscript and has enabled us to improve the manuscript. Each suggestion revision and comment was accurately incorporated and considered. We have read through comments carefully and have made corrections. Based on the instructions provided in your letter, we uploaded the file of the revised manuscript. In order to make the changes easily viewable for you, we used revision mode in the revised manuscript.

  1. Although the authors have synthesized 39 compounds, only the I34 was extensively tested. Leave only the RMN data of I34 in the manuscript; the RMN of the other compounds could be added as supplementary material with their structures.

Reply: We greatly appreciate the reviewer's suggestion. We leave only the RMN data for I34, the RMNs for other compounds have been placed in the supplementary material.

  1. Make an extensive spell check, e.g. lines 24, 61, 99, 101, 132, and so on.

Reply: We apologise for the grammatical errors in the draft and we are very grateful for your careful review. We have checked the format of the text throughout this manuscript and some errors in formatting have been corrected.

  1. Line 72: Correct spelling frugiperda

Reply: Thank you for your careful review, we have corrected the spelling mistake, see in line 74.

  1. Line 122: Provide the concentration of PI and RNAse.

Reply: Thank you for your suggestion, we have added in the text, see in line 125-126.

  1. Line 537: First time that appears C. pipienspallens change by Culex pipienspallens.

Reply: Thank you for your careful review. We have Changed “C. pipienspallensas”to “Culex pipienspallens”.

  1. Line 538: In materials and methods, you wrote that the light distance is 17 cm instead 15 cm. Which is correct?

Reply: We apologise for the errors in the draft. We confirmed that the light distance is 17 cm and have made changes in the manuscript.

  1. Line 592: (C) is a statistic analysis of figure 2E.

Reply: Thank you for your review. We have made changes in the manuscript, see in line 319 .

  1. Line 645: Put the scale bar in figure 3D.

Reply: We are sorry for that the scale bar in figure D 3 is not clear due to the problem in the figure merging process. We upload the original image as supplementary material.

  1. Line 682-683: Confocal microscopy in this magnification is unsuitable for showing morphological changes. I Suggest removing this phrase, as well as the figure 4E. If you wish, can put the figure as supplementary material.

Reply: This is a very good suggestion. Combination with comment 10, we have redrawn figure 4, removing the original fig.4E and adding the statistic analysis of fig.4F. You can see it in fig.4 in revised manuscript.

  1. Line 713: Suggest including the statistic analysis of figure 4F.

Reply: Thank you for your suggestions. we have modified the image based on the comments 9 and 10, you can see it in fig.4F in revised manuscript.

  1. Line 798: The authors wrote that they used UV light for 3 h, but in the materials and methods, the time was 3 min (line 110) which is a huge difference. Please correct.

Reply: In our study, it was actually I34 treatment for 3h followed by 3 min of light. We have also made changes to the corresponding content in the revised manuscript, including.

Kind Regards,

Yanping Luo
